

# Unprecedented atmospheric conditions (1948 – 2019) drive the 2019 exceptional melting season over the Greenland ice sheet

Marco Tedesco[1,2] and Xavier Fettweis[3]

[1]Lamont-Doherty Earth Observatory, Columbia University, Palisades, NY 10964, USA.
[2]NASA Goddard Institute for Space Studies, New York, NY 10025, USA.
[3]Department of Geography, University of Liège, Liège 4000, BELGIUM

*Correspondence to*: Marco Tedesco (mtedesco@ldeo.columbia.edu)

**Abstract.**

Understanding the role of atmospheric circulation anomalies on the surface mass balance of the Greenland ice sheet (GrIS) is fundamental for improving estimates of its current and future contributions to sea level rise. Here, we show, using a combination of remote sensing observations, regional climate model outputs, reanalysis data and artificial neural networks, that unprecedented atmospheric conditions (1948 – 2019) occurring in the summer of 2019 over Greenland promoted new

records or close-to-record values of SMB, runoff and snowfall. Specifically, runoff in 2019 ranked second within the 1948 - 2019 period (after 2012) and first in terms of surface mass balance negative anomaly for the hydrological year September 1, 2018 – August 31, 2019. Summer of 2019 was characterized by an exceptional persistence of anticyclonic conditions that, in conjunction with low albedo associated with reduced snowfall in summer, enhanced the melt-albedo feedback by promoting the absorption of solar radiation and favored advection of warm, moist air along the western portion of the ice sheet towards

the North, where the surface melt has been the highest since 1948. The analysis of the frequency of daily 500 hPa geopotential heights obtained from artificial neural networks shows that the total number of days with the most frequent atmospheric pattern that characterized the summer of 2019 was 5 standard deviations above the 1981 – 2010 mean, confirming the exceptional nature of the 2019 season over Greenland.

## 1    Introduction

Understanding the role of atmospheric circulation changes on the surface mass balance (SMB) of the Greenland ice sheet (GrIS) is crucial for improving estimates of its current and future contribution to sea level changes and for studying recent mass loss trends in the context of multi-decadal timescales. Atmospheric patterns modulate the GrIS mass balance through snowfall and runoff (e.g., Hanna et al., 2008, 2013, 2016; Tedesco et al., 2011, 2016a, 2016b) as well as radiative forcing and surface turbulent heat fluxes (e.g., clouds, longwave and shortwave radiation). Recent studies (e.g., Hanna et al.,

2014; Mattingly et al., 2016; McLeod and Mote, 2016) have focused on linking the observed variability of climate indices





such as the North Atlantic Oscillation (NAO, i.e., Hanna et al., 2015) or the Greenland Blocking Index (GBI, Hanna et al., 2018) to the recent changes in runoff and accumulation over Greenland. Other studies (i.e., Tedesco et al., 2016b) have recently pointed out to the increased frequency of persistent anticyclonic conditions favoring atmospheric blocking and explaining most of the recent surface melt increase (Fettweis et al., 2013).

35        In this paper, we report the results of an analysis of SMB and surface energy balance (SEB) components obtained from satellite data and model outputs for the summer of 2019, their linkages to anomalies in the atmospheric circulation and analyze them within the long-term context (1948 – 2019). Specifically, we use spaceborne passive microwave data collected between 1979 and 2019 at 19.35 GHz, horizontal polarization, for detecting melting following the approach reported in Tedesco, 2007; Tedesco et al., 2007 and Tedesco, 2009). We also use estimates of broadband albedo derived from data

collected by the Moderate Resolution Imaging Spectroradiometer (MODIS) for the period 2000 – 2019 (https://terra.nasa.gov/about/terra-instruments/modis). We complement satellite data with the outputs of the Modèle Atmosphérique Régionale (MAR) regional climate model (RCM, Gallée and Schayes, 1994; Gallée, 1997; Lefebre et al., 2003) forced by National Centers for Environmental Prediction/National Center for Atmospheric Research (NCEP-NCARv1, Kalnay et al., 1996) reanalysis dataset over the period 1948 – 2019. We lastly make use of self-organizing maps

(SOMs, i.e. ,Kohonen 2001) to classify pan-Arctic summer 500 hPa geopotential height (GPH) anomalies (1981 – 2010 baseline period) also obtained from the NCEP/NCAR reanalysis dataset (Kalnay et al., 1996) between 1948 and 2019. The pan-arctic region is here defined as the portion of the northern hemisphere poleward of 60º N. We focus on the 500 hPa GPH values because of their strong correlation with SMB quantities and for consistency with other studies using them to compute climate indices, such as the GBI (e.g. Hanna et al., 2016). Moreover, 500 hPa is also a standard height for gauging the effects

of jet stream blocking on synoptic weather patterns (e.g. McIlveen, 2010).

## 2    Methods and data

### 2.1    Satellite data

      Passive microwave (PMW) brightness temperatures (Tbs) are a crucial tool for studying the evolution of melting over the Greenland and Antarctica ice sheets (e.g. Abdalati and Steffen, 1995; Tedesco, 2007; Tedesco et al. 2009; Tedesco

2009; Fettweis et al., 2011). The capability of passive microwave sensors to collect useful data during both day- and nigh-time and in all-weather conditions provides data at a high temporal resolution (at least daily over most of the Earth), with high latitudes being covered several times during a single day. Since the launch of the Scanning Multichannel Microwave Radiometer (SMMR) in October 1978, Tb data is available in multiple bands every other day (in the case of SMMR) and daily starting in 1987, with the launch of the Special Sensor Microwave Imager (SSMI). PMW brightness temperature

records are the longest available time series and an irreplaceable tool in climatological and hydrological studies, especially for those regions, such as the ice sheets, where in-situ observations lack and fieldwork is logistically difficult, if not impossible. Specifically, we make use of data distributed by the National Snow and Ice Data Center (NSIDC,



https://nsidc.org/; https://catalog.data.gov/dataset/near-real-time-dmsp-ssm-i-ssmis-pathfinder-daily-ease-grid-brightness-temperatures-version) at a spatial resolution of 25 km at K band (~ 19 GHz), horizontal polarization. Melting is detected
following the procedure described in Tedesco (2007) and Tedesco (2009).

We complement PMW data with the MODIS daily surface reflectance product (MOD09GA Version 6) and daily snow cover product (MOD10A1 Version 6, https://nsidc.org/sites/nsidc.org/files/files/MODIS-snow-user-guide-C6.pdf). The MOD10A1 data include broadband albedo estimated based on the MOD09GA product. We used the Version 6 data in view of its improvement in sensor calibration, cloud detection, and aerosol retrieval and correction relative to version 5 (e.g.,
Casey et al., 2017). Version 6 data are optimal for assessing temporal variability of surface albedo as they are corrected for sensor degradation issues impacting earlier versions (Casey et al., 2017). The spatial resolution of the MODIS datasets is 500 m. We use the cloud mask in the MOD10A1 data to exclude clouds.

## 2.2    The MAR regional climate model

The regional climate model MAR (Fettweis et al., 2017) combines atmospheric modeling (Gallée and Schayes,
1994) with the Soil Ice Snow Vegetation Atmosphere Transfer Scheme (De Ridder and Gallée, 1998) and has been extensively evaluated and used to simulate surface energy balance and mass balance processes over GrIS (.eg. Fettweis, 2007; Fettweis et al., 2011); . In this study, we use the version 3.10 of MAR, at a horizontal spatial resolution of 20km as in Fettweis et al. (2017) and 6 hour temporal resolution forced with the NCEP-NCARv1 reanalysis (Kalnay et al., 1996). Outputs generated at sub-daily temporal resolution are, then, averaged to obtain daily values. We refer to Fettweis et al.
(2017) for the evaluation of this NCEP-NCARv1 forced simulation and to Delhasse et al. (2019) for the list of improvements made since MARv3.5 used in Fettweis et al. (2017).

## 2.3    NCEP/NCAR reanalysis data and the Greenland Blocking Index (GBI)

We use geopotential heights at 500 hPa obtained from the NCEP/NCAR reanalysis dataset, consisting of globally gridded data that incorporate observations and outputs from a numerical weather prediction model from 1948 to present
(Kalnay et al., 1996). We also use the so-called Greenland Blocking Index (GBI), defined as the mean 500 hPa geopotential height over the area bounded by the following coordinates 60-80°N, 20-80°W (e.g., Hanna et al. 2015, 2018). Positive GBI conditions are generally associated with surface high pressure 'blocking' anomalies over the Greenland region (Hanna et al., 2016). There is also a strong and significant anti-correlation between Greenland blocking and the North Atlantic Oscillation (NAO, the first mode of atmospheric surface pressure variation over the North Atlantic), with Greenland blocking typically
linked to a southward deflection of the jet stream (Hanna et al. 2015 ,2018; Tedesco et al., 2016b). Here, we use a recent reconstruction of GBI from 1851-2019 (Hanna et al., 2018) that combines data from the 20CRV2c Reanalysis (Compo et al., 2011) with newer (1948-2015) data from the NCEP/NCAR reanalysis (Kalnay et al. 1996). The overall integrity of the long-term GBI time series is ensured by using homogeneity adjustments (Hanna et al. 2016).



## 3     Results

Melt duration in 2019 (**Fig. 1a**) estimated from PMW data exceeded the long-term (1981 - 2010) mean by up to 40 days along the west portion of the ice sheet where dark, bare ice is exposed (**Fig. 1b**). Over the rest of the ice sheet, the anomaly of the number of melting days during the summer of 2019 from PMW data was around 20 days. Negative anomalies were rare and geographically concentrated over a small area in the southern portion of the ice sheet. Surface melting in 2019 started relatively early, around mid April (**Fig. 2a**), and exceeded the 1981 – 2010 mean for ~ 82 % of the

days during the period June 1 – August 31, 2019. The melting index in 2019 (MI, defined as the number of melting days times the area undergoing melting and being a measure of the intensity of surface melting, i.e., Tedesco, 2007) ranked second, after 2012. When looking at the different summer months separately, the MI values in 2019 ranked 5[th] in June, 7[th] in July and 9[th] in August (**Fig. 2b**). The 2019 updated trends for MI and melt extent (here defined as the area subject to at least one day of melting) are, respectively, 78.836 $Km^2$*day decade$^{-1}$ (p<<0.01, MI) and 7.66 % decade$^{-1}$ (p<<0.01, trend is here

expressed as a percentage of the total area of the ice sheet). The maximum daily melt extent was reached on July 31, 2019 covering ~ 73 % of the ice sheet surface. In comparison, the average daily maximum extent from PMW data for the same day for the 1981–2010 period is 39.8%. Notably, the total area that at anytime underwent melting was 95.8 % of the total ice sheet in 2019 (**Fig. 2b**), against the 1981 - 2010 averaged value of 64.3 %. Indeed, the same atmospheric event at the end of July that was responsible for promoting melting over 73 % of the ice sheet surface was also responsible for the cumulative 3-

day melt event that extended over high elevations and that covered up to ~ 96% of the ice sheet surface. We note that a similar value for the maximum melt extent was reached in 2012, though in this case it did happen in one day. As in 2019, the exceptional melt in 2012 was associated with the advection of a very warm and wet air masses coming from the south and promoting the presence of liquid water clouds promoting surface melt in the dry snow zone. However, in 2019, the air mass came from the east after promoting an exceptional heat wave in Europe, being warmer and drier than the air mass in 2012.

Moreover, by crossing the relative cold Atlantic Ocean from Scandinavia, in 2019 the lower atmospheric layers cooled down increasing the stability of the air mass and then limiting the formation of liquid water clouds compared to July 2012, explaining why the melt extent was lower during this 2019 big melt event than in July 2012 while the temperature anomaly was higher in the free atmosphere in 2019 than in 2012.

We investigated the possibility that the sporadic melting detected at high elevations could have been due to a

malfunctioning of the sensor or other issues related to data quality. **Fig. 3a** shows a map of the number of melting days constrained to values ranging between 1 and 4 days to highlight those areas where melting occurred for a few days at high elevations. In the figure, we also show the time series of brightness temperatures for those pixels where melting occurred for only one day (**Fig. 3b**) or for 2 days (**Fig. 3c**). The sharp, sudden increase of brightness temperatures is not associated with data quality issues but rather with the insurgence of melting in both cases. Melting at high elevations is also confirmed from

the analysis of in-situ data. For example, **Fig. 4a** shows air temperature (2m) recorded at the EGP PROMICE station (75.6247°N, 35.9748°W, 2660 m a.s.l., https://www.promice.dk/WeatherStations.html) together with time series of



spaceborne Tbs at 19.35 GHz, horizontal polarization, recorded over the pixel containing the location of the EGP station (blue line). Air (2m) pressure (hPa) recorded at the same station is also reported as a red line in the bottom plot. The figure shows that air temperature exceeded the value of 0ºC when Tb values sharply increased from ~ 170 k to ~ 220k.

Concurrently, surface air pressure reached at EGP peak values of ~ 749 hPa, likely as a consequence of the persistent anticyclonic conditions occurring during that period.  We also note that air temperature exceeded the melting point at least twice in 2019 at the EGP station beside July 31, according to the in-situ data. The first time on day 163 (June 12) and the second time on day 201 (July 19). In both cases, however, the passive microwave data did not detect the presence of liquid water. This might be a consequence of the fact that air temperature can be exceeding the melting point when snow

temperature is not and that the second event when air temperatures exceed the melting point was characterized by relatively low pressure, hence suggesting that the radiative forcing associated with the incoming solar radiation might have not been as strong as in the case of the end of July.

The spatial distribution of the anomaly of the number of melting days obtained from PMW observations is consistent with the one obtained from the MAR regional model, as shown in **Fig. 5a**. Here, we consider those cases when the

integrated liquid water content in the top meter of the snowpack reaches or exceeds 1 mmWE, following Fettweis et al. (2007). Meltwater runoff in JJA 2019 simulated by MAR and integrated over the whole ice sheet ranked second (consistently with the MI values obtained from the PMW data), reaching a total of 560 Gt in 2019 against an average value of 300±85 GT yr$^{-1}$ for the 1981-2010 period**.** As a reference, the value of runoff simulated by MAR for the JJA 2012 period (when the record was established) was 610 Gt. Despite ranking second in terms of surface runoff, the September 2018 -

August 2019 (used to define the mass balance "year") ranks first in terms of integrated SMB negative anomaly simulated by MAR, with a total surface mass loss anomaly of ~ 320 Gt yr$^{-1}$ with respect to the 1981-2010 SMB average, breaking the previous record established in 2011-2012 of ~ 310 Gt yr$^{-1}$ (**Fig. 6,** blue bars**)**, though by only 10 Gt yr$^{-1}$. It is however important to note that such difference is below the uncertainty of the MAR model estimated to be 10% of the mean SMB.

The SMB negative anomaly in 2018-2019 is larger than 2011-2012 mainly because the 2018-2019 snowfall

negative anomaly (~ - 50 Gt) is larger in magnitude than the one that occurred during the 2011-2012 SMB year (~ -20 Gt), with large negative summer snowfall anomalies in 2019 occurring along the southern and western portions of the ice sheet (**Fig. 5b)**. The early melt onset and the negative snowfall anomaly promoted the exposure of bare ice prematurely, hence further enhancing melting and runoff through the melt-albedo positive feedback mechanism (i.e., Tedesco et al., 2016b). This is evident from the analysis of summer broadband albedo simulated by MAR (**Fig. 5c)**, showing negative anomalies

down to -0.2 along the western portion of the ice sheet. These results are also confirmed by albedo estimates obtained from MODIS (**Fig. 7a**), indicating a large, negative albedo anomaly occurring along the west coast where bare ice is exposed. Specifically, summer MODIS albedo ranked 4th (**Fig. 8**) within the 2000 – 2019 MODIS period, being -1.45 standard deviations (σ) below the mean (2000 – 2010 baseline period). The summer of 2019 precedes the ones of 2010 (-1.79σ), 2016 (-1.95σ) and 2012 (-3.33σ) in terms of MODIS albedo. When considering the summer months separately, June and July

2019 ranked, respectively, 10$^{th}$ (June) and 7$^{th}$ (July). A new record was, nevertheless, established in August 2019, with the





absolute value averaged over the whole ice sheet reaching 77.51 % (-2.39σ) in 2019, followed by 2012 (77.86 %, -2.05σ) and 2016 (78.1 %, -1.81σ). The updated trends over 2000-2019 for summer broadband albedo is -0.4% decade$^{-1}$, though it is not statistically significant (R$^2$ = 0.04). Similarly, the trends for June (-0.1 %), July (-0.6 %) and August (-0.7 %) are also not statistically significant.

165         The analysis of the maps of the monthly averaged albedo (**Fig. 7 b through d**) indicates that, as mentioned above, negative albedo anomalies occurred along the western portion of the ice sheet in June and July but, during the same period, albedo was within the average over most of the rest of the ice sheet. In June, only 23 % of the ice sheet surface was showing positive albedo anomalies was ∼ 23 %. The value for July was 25 %, to be reduced to only 6 % in August. During this month, the negative albedo anomalies in the south are confined along a relatively small portion of the west margin of the ice
sheet, but they extend further inland, reaching high elevations in the northern regions (**Fig. 7c**). The presence of negative albedo anomalies in August at higher elevations is consistent with the sporadic melting that occurred over the same region at the end of July and beginning of August 2019 (**Fig. 3**). The impact of such event is, indeed, well observable in the albedo changes of the pixel that underwent melting for two days at the end of July (**Fig. 9,** being the same as the one whose Tb values are shown in **Fig. 3b**), showing a reduction from 87.4 % to 77.8 % due to the increase in grain size associated with the
melting and refreezing cycle.

## 4    Discussion

        A major driver of the exceptional melting season in 2019 was the persistency of high pressure systems over the GrIS that promoted an increase in the absorbed solar radiation as well as the flow of warm, moist air along the western portion of the ice sheet towards the north of the ice sheet. The anticyclonic conditions were responsible also for reduced
cloudiness in the south and consequent below-average summer snowfall and albedo in this area. Similarly to 2012, anticyclonic conditions dominated summertime (**Fig. 10a**). The anomaly also occurred at the surface (**Fig. 10b**), suggesting that the pressure anomaly in the mid-troposphere was driven by atmospheric circulation rather than by the warming of the free atmosphere below 500 hPa levels. The anticyclonic conditions also promoted the advection of warm air that reached the northern portion of the ice sheet explaining why the highest temperature anomaly at 700 hPa occurs in this area (**Fig.10c**).
Over the center of the ice sheet, surface temperature was close to the 1981 – 2010 average, suggesting a larger role of the radiative forcing than the thermal one. The mean summer sea level pressure (SLP) averaged over the 60-80°N, 20-80°W region (i.e., the same area used to compute GBI, Hanna et al., 2016), reached a breaking record value of 1016 hPa vs. a 1981 - 2010 summer average of 1010+/-2 hPa. Also the summer averaged 500 hPa geopotential height anomalies, integrated over the same area, set a new record of 5567 m, against a 1981 - 2010 average of 5497 +/- 25m (**Fig. 11a**). We computed the
persistency of anticyclonic conditions, defined here as the number of days when the daily mean SLP averaged over the Greenland ice sheet exceeds 1013 hPa (the common value of the standard pressure), and found that during the summer of





2019 such conditions existed for 63 of the 92 summer days (68 % of the summer). In perspective, the average number of days with the same conditions during the period 1981 - 2010 was 28+/- 12 days.

The anticyclonic conditions that characterized the summer of 2019 promoted negative cloudiness anomalies over the southern portion of the ice sheet and positive ones over the northern region (**Fig. 5d**), pointing to the important role of clouds in enhancing melting in this area (i.e., Hofer et al. 2017). In the North, the exceptional persistence of a high pressure system centered near Summit over the whole 2019 summer (**Fig. 5a**) favored advection of warm and wet air along the west side of Greenland towards the North, promoting higher than average surface temperatures (**Fig. 5d**) and positive anomalies of long wave downwelling radiation (**Fig. 5f**). In the southwest, dry and sunny conditions dominated. This promoted positive anomalies of the incoming shortwave radiation (**Fig. 5g**) which, in turn, when combined with the relatively low albedo (due to reduced summer snowfall) promoted positive anomalies of the absorbed shortwave radiation (**Fig. 5h**) higher than 30 W $m^{-2}$. Such drier conditions also allowed temperatures to reduce during nighttime, explaining why the temperature anomaly was not playing a larger role over these regions. Integrated over the whole ice sheet, the anomalies of shortwave and long wave downwelling radiation were not significant but, as a result of a quasi permanence of exposure of low albedo zones, the anomaly of absorbed shortwave was the highest since 1948, with an anomaly integrated over the whole ice sheet of 7.9 W $m^{-2}$, being four times the 1981-2010 standard deviation (inter-annual variability) of 1.9 W $m^{-2}$. The strong relationship between runoff and atmospheric conditions is also apparent in **Fig. 12,** where scatter plots of runoff with 500 hPa GPH summer mean anomalies (**Fig. 12a**) and with 700 hPa temperature (**Fig. 12b**) are shown, together with the coefficients of the linear regression between runoff and the two atmospheric quantities. Reinforcing the idea that radiative forcing played a large role with respect to thermal forcing, the summer of 2019 (marked in the two panels with a large, orange circle) is beyond two standard deviations from the mean in the case of the 700 hPa temperature where it falls closely to the regression line in the case of the 500 hPa GPH.

To further understand the role of the atmosphere on the 2019 SMB record and the linkages between atmospheric circulation and SMB, we classified summer (JJA) daily 500 hPa GPH between 1948 and 2019 into a set number of classes to study how the frequency of such classes has changed over the past decades and how the 2019 summer positioned itself within the 1948 – 2019 record. We focus on the 500 hPa GPH because of its strong correlation with the surface melt (Fettweis et al., 2011b) and because it is a standard height for gauging the effects of jet stream blocking on synoptic weather patterns (e.g. McIlveen, 2010). We classify the daily 500 hPa GPH by means of Self Organizing Maps (SOMs), being artificial neural network algorithms that use unsupervised classification to perform nonlinear mapping of high-dimensional datasets (Kohonen 2001). During the training phase of the SOMs, each of the daily 500 hPa GPH fields is allocated to one of the classes depending on the Euclidean distance of the new element from the existing SOM nodes. Once trained, the SOM network is interrogated by providing the daily 500 hPa GPH anomaly fields (1981 – 2010 baseline) and obtaining the corresponding class to which that particular atmospheric field belongs. From here, it is possible to calculate the frequency of occurrence of the classes of the atmospheric circulation patterns to provide insight into possible temporal changes associated with the identified classes and their relationship with SEB and SMB quantities. The number of nodes, which also



corresponds to the number of classes in which the atmospheric patterns are classified (Kohonen, 2001), is defined by the user: using fewer nodes allows the user to include a broader range of circulation patterns within the same class but it decreases the amount of variability captured by the SOMs, while increasing the number of nodes results in classes that are less frequent and more closely resemble each other. Based on previous work (e.g., Mioduszewski et al., 2016) and following

Kohonen (2011), we selected a total number of 28 classes. **Fig. 13** shows the 28 nodes identified through the SOM analysis ordered according to the mean GPH values computed over the same area where GBI is calculated. For each node, the position of each class in the original grid is reported (shown as class #) together with the mean 500 hPa GPH value. The maps in **Fig. 13** are obtained by averaging the 500 hPa GPH values over those days when the specific node was occurring according to the SOM classification. For reader's convenience, in **Fig. 14a** we show the anomaly of the summer frequency

of occurrence of each class (y-axis) for the years 1949 through 2019 (x-axis) with respect to the 1981 – 2010 period. Further, in **Fig. 14b** we show the number of days occurring in 2019 (blue bars) and 2012 (red line) for the different classes (x-axis). We selected 2012 and 2019 because of the enhanced surface melting that characterized both summers. We note that the atmospheric patterns characterizing the two summers show differences and similarities. Both summers, indeed, had a high number of days when class # 20 (highlighted with a rectangle with dashed contour in **Fig. 13)** was occurring (up to ~ 10 days

in 2019). This class is characterized by large positive 500 hPa GPH anomalies (above 80 m) over Greenland and the Canadian archipelago, negative anomalies over Scandinavia and large positive anomalies over Siberia. Differently from 2012, however, classes # 11, 12, 13 and 28 were persistently present in 2019 (highlighted in **Fig. 13** with rectangle with a continuous line). Classes # 12 and #13 show relatively low 500 hPa GPH anomalies over Greenland but strong positive anomalies over the Arctic ocean (class # 13) and the Canadian archipelago, eastern Siberia and Scandinavia (Class # 12).

Classes # 11 and 28 show large positive anomalies over Greenland reaching both the Canadian archipelago and northern Europe and relatively high positive 500 hPa GPH anomalies over Siberia and Alaska. Notably, the cumulative number of days for classes # 11, 12, 13, 20 and 28 above identified exceeded 55 days in 2019 (**Fig. 14c)** being 5.1 standard deviations above the 1981 – 2010 mean of 14.2 days and pointing out, again, to the exceptional nature of the atmospheric conditions over Greenland during the summer of 2019.

**5    Conclusions**

Using a combination of remote sensing observations, regional climate model outputs and reanalysis datasets as well as self organizing maps (SOMs), we have shown that exceptional anticyclonic conditions occurred in the summer of 2019 that promoted new records or close-to-record values of SMB, runoff and snowfall. Runoff in 2019 was the second highest after 2012 and SMB was the lowest on the record according to MAR forced by NCEP-NCARv1. The exceptional nature of

the mass balance components in 2019 was strongly driven by albedo reduction associated with reduced summer snowfall, enhanced absorption of solar radiation and the flow of warm, moist air along the western portion of the ice sheet. The analysis of the frequency of daily 500 hPa GPH obtained from SOMs shows that the persistency of the atmospheric patterns





(i.e., frequency expressed as number of days) characterizing most of the 2019 summer was unprecedented, being 5 standard deviations above the 1981 – 2010 mean, confirming the exceptional nature of the 2019 season over Greenland. Despite similar in terms of runoff and SMB, the 2012 and 2019 exceptional melting seasons differ in terms of atmospheric patterns that drove those exceptional conditions, highlighting the importance of studying the spatio-temporal evolution of the atmospheric quantities, rather than only looking at integrated indices such as NAO ad GBI. In the future, we plan to analyse how the frequency and occurrence of the atmosphere has been changing at higher geopotential height levels (e.g., 300 hPa, 100 hPa) to eventually quantify potential missing links between the stratosphere and the troposphere that might be responsible for the exceptional conditions. We plan to look at these potential linkages during the fall and winter months, when the coupling between the stratosphere and the troposphere is stronger than in summer and will explore the potential influence of winter and spring conditions on the summer atmosphere. As mentioned in the Introduction, understanding the role of atmospheric circulation changes on the surface mass balance of the Greenland ice sheet is a crucial step for improving estimates of its current and future contributions to sea level changes. This assumes even more importance when considering that such exceptional conditions are not captured by the Climate Model Intercomparison Project datasets (CMIP5, Hanna et al., 2018b), and they can increase the projected surface mass loss by a factor 2 according to Delhasse et al. (2018) .

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



## 7    Acknowledgements

M. Tedesco would like to acknowledge financial support by the National Science Foundation (PLR-1603331, PLR-1713072, OPP 19-01603), NASA (NNX17AH04G, 80NSSC17K0351) and the Heising-Simons foundation. Computational resources

for running MAR have been provided by the Consortium des Équipements de Calcul Intensif (CÉCI), funded by the Fonds de la Recherche Scientifique de Belgique (F.R.S.FNRS) under grant no. 2.5020.11 and the Tier-1 supercomputer (Zenobe) of the Fédération Wallonie Bruxelles infrastructure funded by the Wallonia region under the grant agreement no. 1117545.

**Data availability**


All        MARv3.10        outputs        presented        here        are        available        on
ftp://ftp.climato.be/fettweis/MARv3.10/Greenland/NCEP1_1948-2019_20km/. Remote sensing data are available at        the        links        mentioned        within        the        text.        The        MAR        code        is        available        at
http://mar.cnrs.fr/index.php?option_smdi=presentation&idm=10.


**Competing interests.**

The authors declare they have no conflict of interest.

**Authors contribution.**

MT and XF conceived the study. MT collected and analyzed the remote sensing data and performed the atmospheric classification using the SOMs. XF generated the MAR outputs. Both authors contributed to the analysis and to the final version of the manuscript.



**Figures**

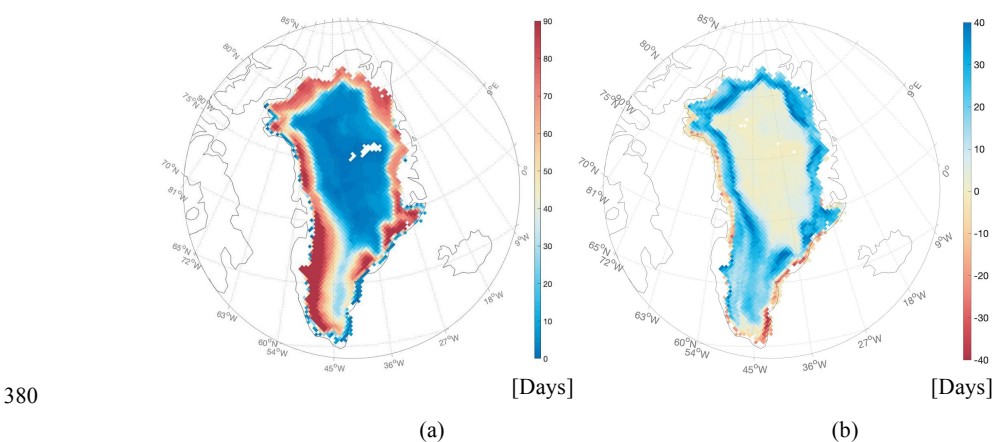


        (a)                                           (b)

**Figure 1 a) Number of days when melting occurred during the 2019 summer (June, July, August, JJA) according to spaceborne passive microwave observations (e.g., Tedesco et al., 2007). b) Anomaly of the number of melting days with respect to the 1981 – 385  2010 baseline period obtained from spaceborne passive microwave data shown in a).**

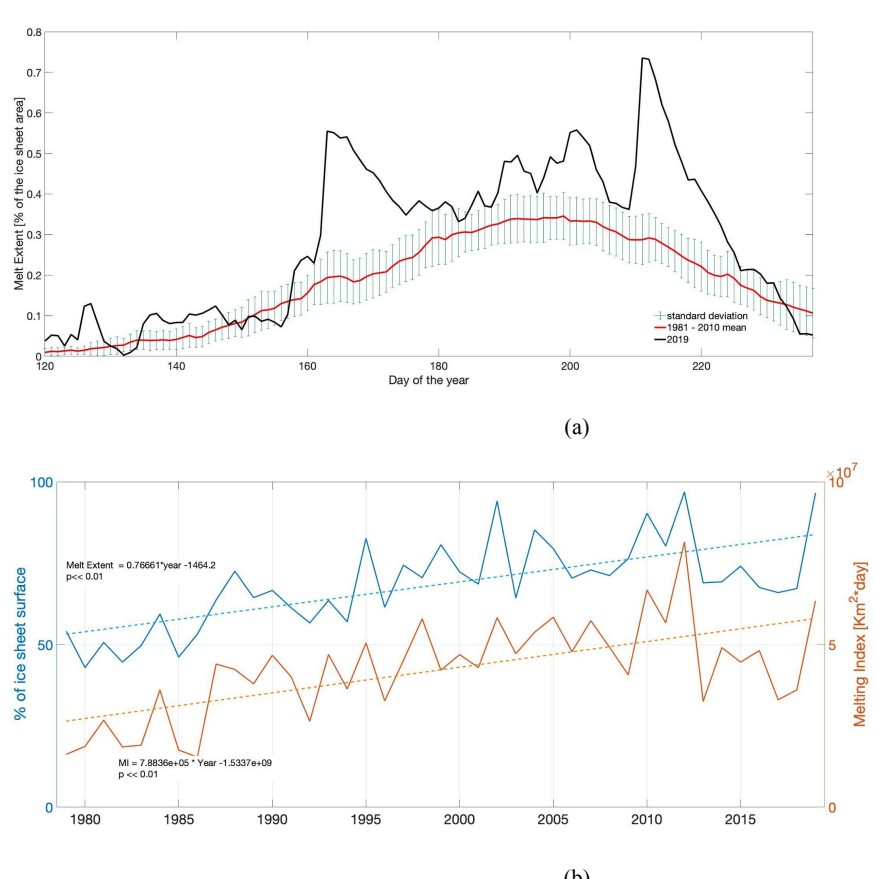

(a)


(b)

**Figure 2 a) Daily time series of melt extent (expressed as a percentage of the total ice sheet area) during 2019 (black line). Red line indicates the average values for the baseline period 1981 – 2010. Vertical gray bars indicate the standard deviation of melt extent**

**for the 1981 – 2010 baseline period. b) Summer-averaged melt extent (as a percentage of the ice sheet surface, blue line, left axis) and melting index (e.g., number of melting days times the area undergoing melting, Km²*day, orange line, right axis) obtained from spaceborne passive microwave observations for the period 1979 – 2019.**





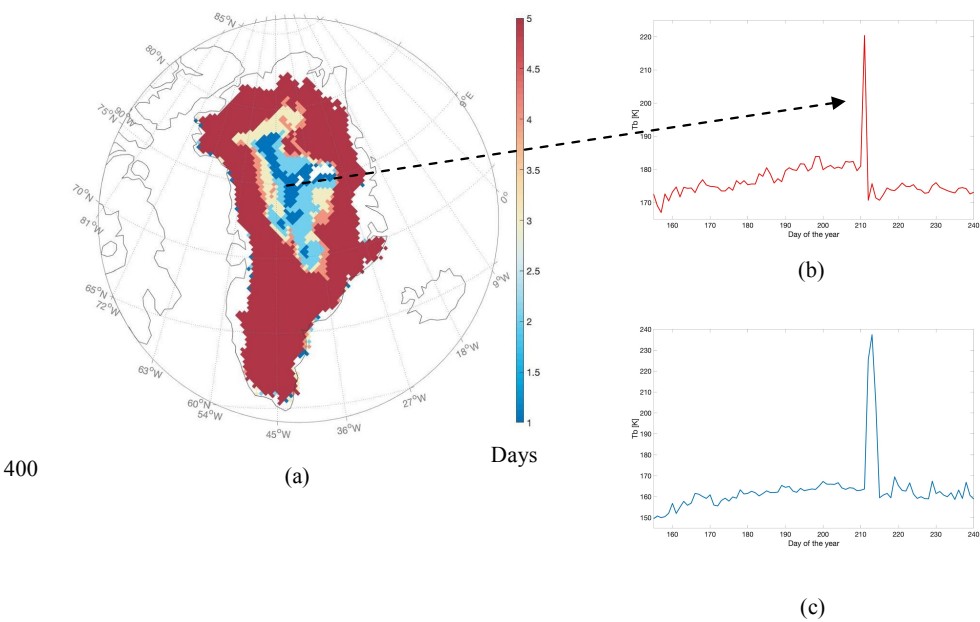

400                                     (a)

Days

(b)

(c)

Figure 3 a) Number of melting days in 2019 obtained from spaceborne passive microwave observations. The data is the same as Figure 1 but the range has been reduced between 1 and 4 days to highlight melting occurring in the interior at high elevations. b) and c) Daily time series of spaceborne microwave brightness temperatures for the two selected points indicated by the tail of the arrow.






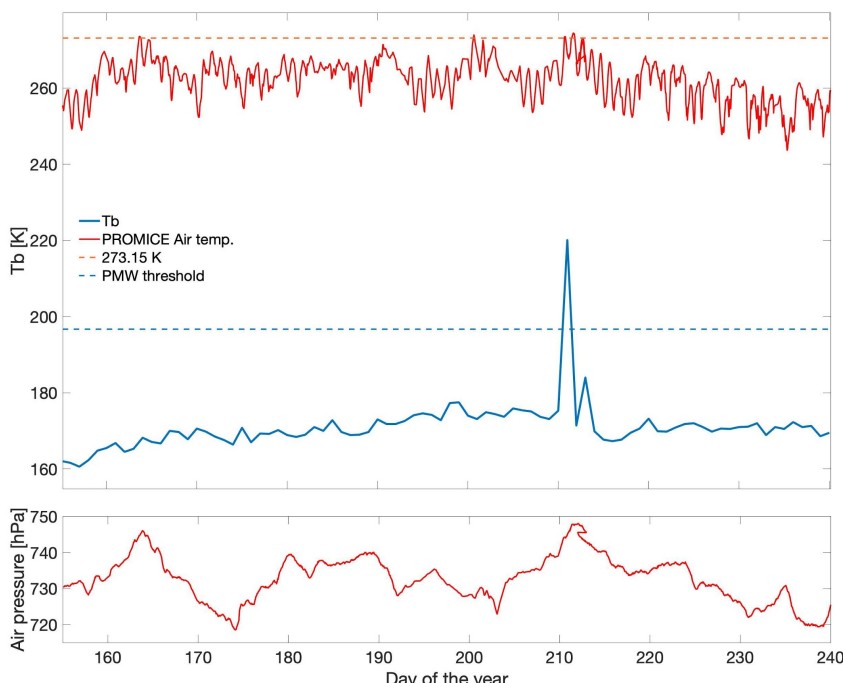

**Figure 4 Time series of daily air temperature (top red line) recorded at the EGP PROMICE station (75.6247ºN, 35.9748ºW, 2660 m a.s.l.) together with time series of spaceborne brightness temperatures at 19.35 GHz recorded over the pixel containing the location of the EGP station (blue line) together with air pressure [hPa] recorded at the same station (bottom red line). Dashed orange line represents the 273.15 K values where the blue, dashed line represents the threshold on Tb above which melting is considered to occur.**

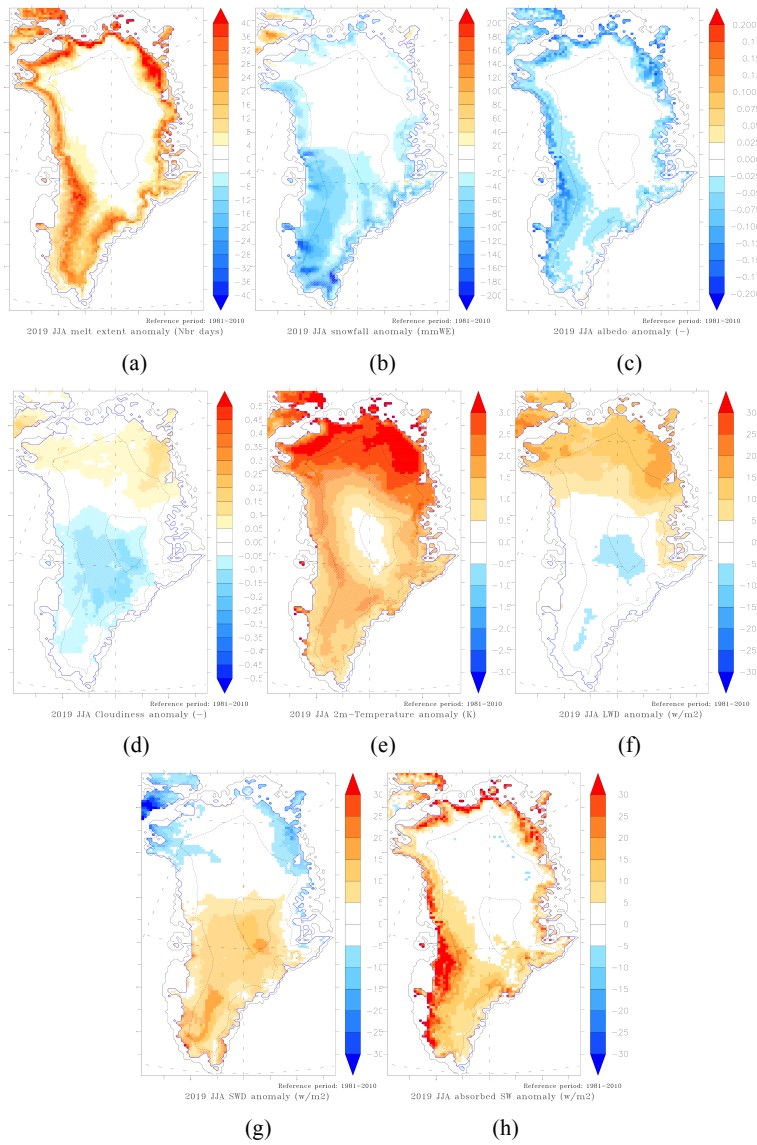


**Figure 5 Spatial distribution of the anomaly of the a) number of melting days b) 2m temperature c) snowfall d) albedo e) cloudiness f) longwave downwelling g) shortwave downwelling and h) shortwave absorbed obtained from the MAR model (1981 – 2010 baseline) forced by the reanalysis NCEP-NCARv1.**




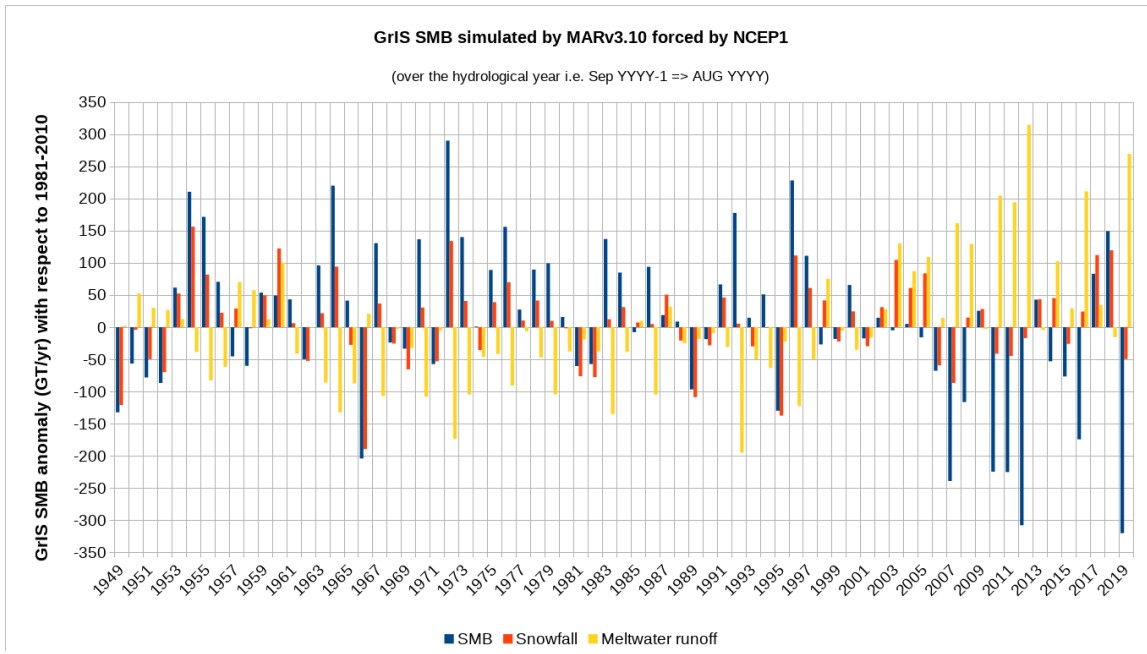

**Figure 6 Time series of 1949 – 2019 annual (Sept. 1 2018 – august 31st, 2019) SMB (dark blue), snowfall (red) and runoff (yellow) values simulated by MAR over the whole Greenland ice sheet.**









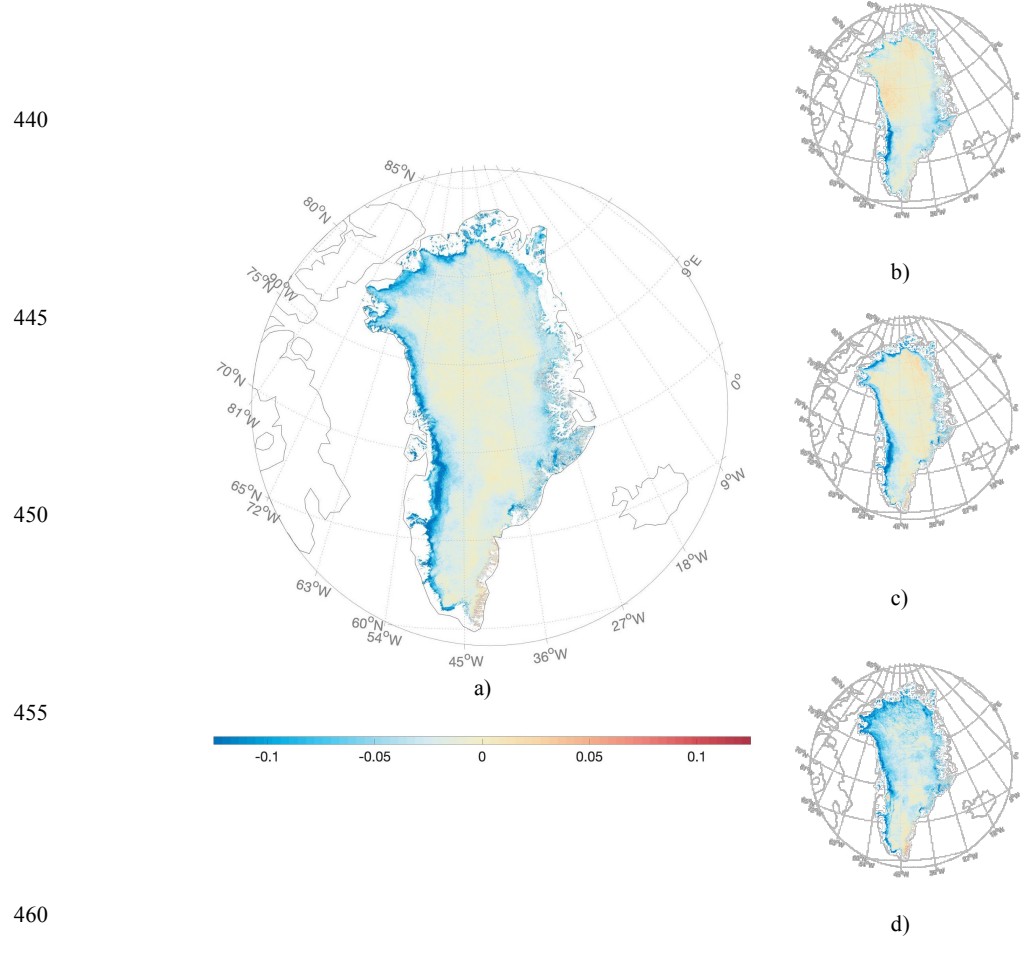

**Figure 7 MODIS 2019 broadband albedo values anomalies (2000 – 2010 baseline) for a) summer b) June c) July d) August.**



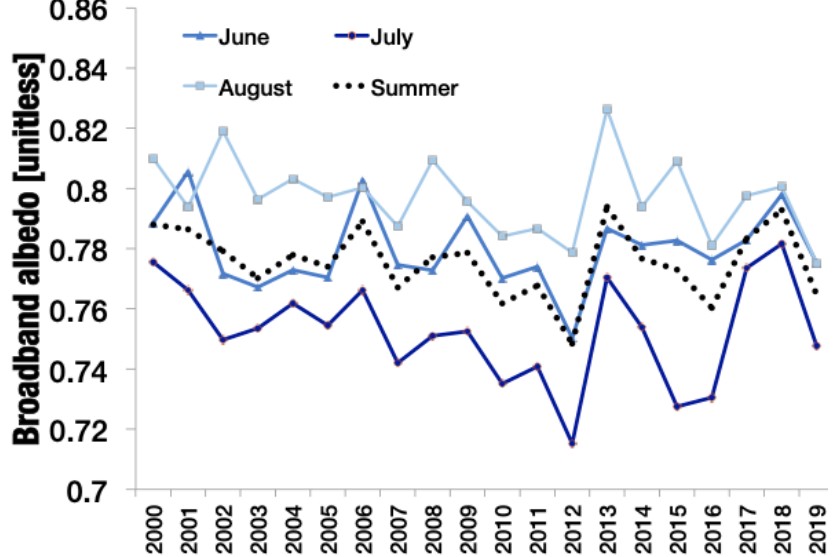

**Figure 8 Time series of MODIS 2019 broadband albedo values for summer (dark, dotted line), June (medium blue line with triangles) July (dark blue line with disks) and August (light blue line with squares).**





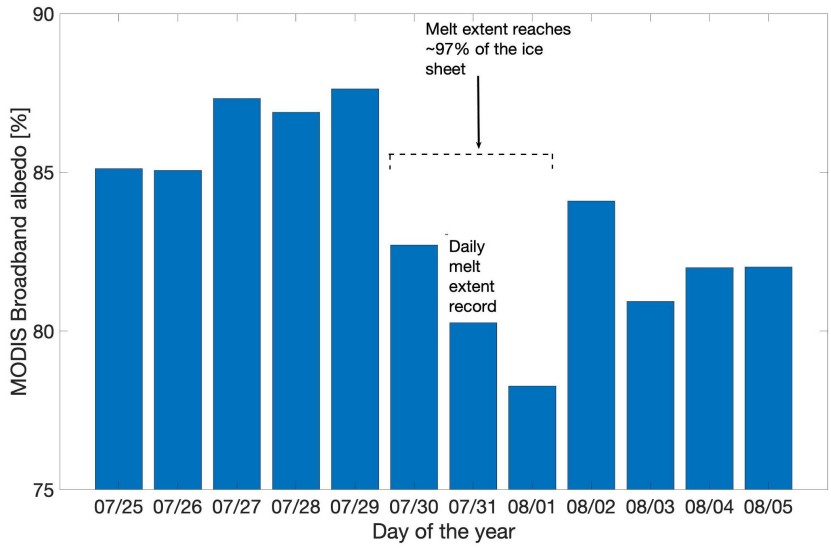

**Figure 9 Time series of values of MODIS broadband albedo for the pixel whose brightness temperature is shown in Fig. 3b.**

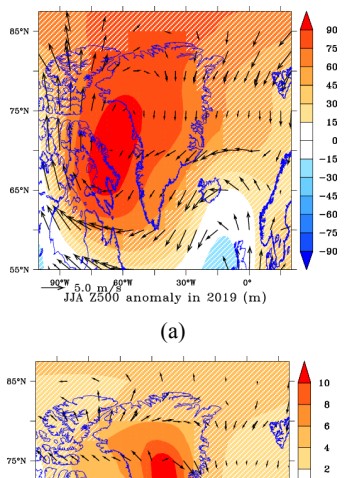

(a)

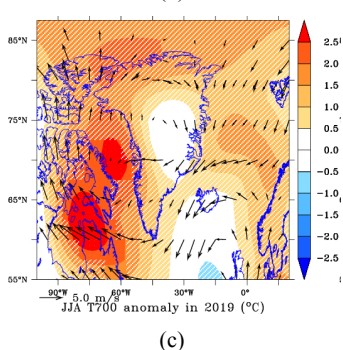

(b)

(c)


**Figure 10 a) Anomaly of the JJA 2019 averaged geopotential height anomalies at 500hPa (Z500 in m, baseline period 1981 - 2010) from the NCEP-NCARv1 reanalysis. Anomalies below two times the interannual variability (i.e. not statistically significant) are hatched. b) Same as a) but for the sea level Pressure (hPa) and for c) JJA temperature at 700hPa (°C). In each panel, arrows represent the anomaly of JJA winds at a) 500 hPa, b) 10 m and c) 700hPa.**






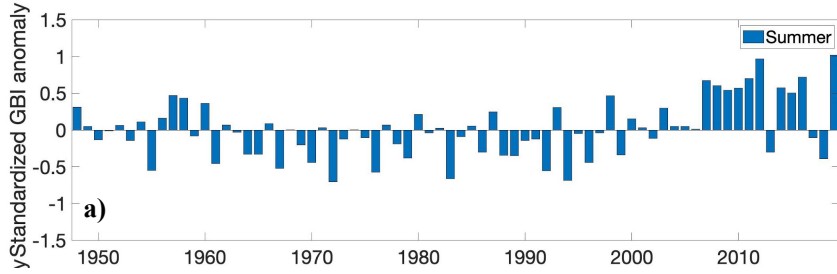

**Figure 11 Standardized (1981 – 2010) a) summer (JJA) averaged GBI values for the period 1948 – 2019.**




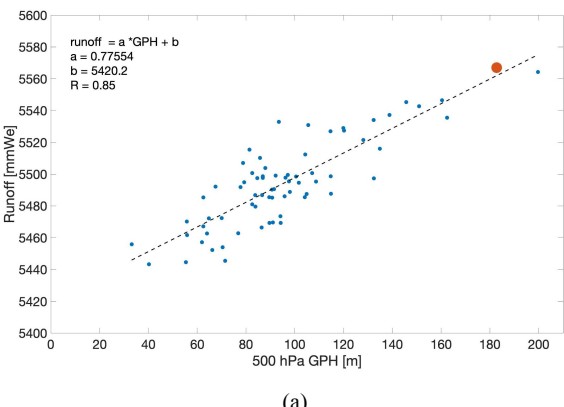

(a)

(b)

**Figure 12 Scatterplot between runoff (in mmWE) and a) 500 hPa GPH (m) and b) 700 hPa Temperature (in ºC). Red disks show**
**2019 values in both plots. The coefficient of a linear regression analysis are reported within each plot, together with the coefficient of determination R.**

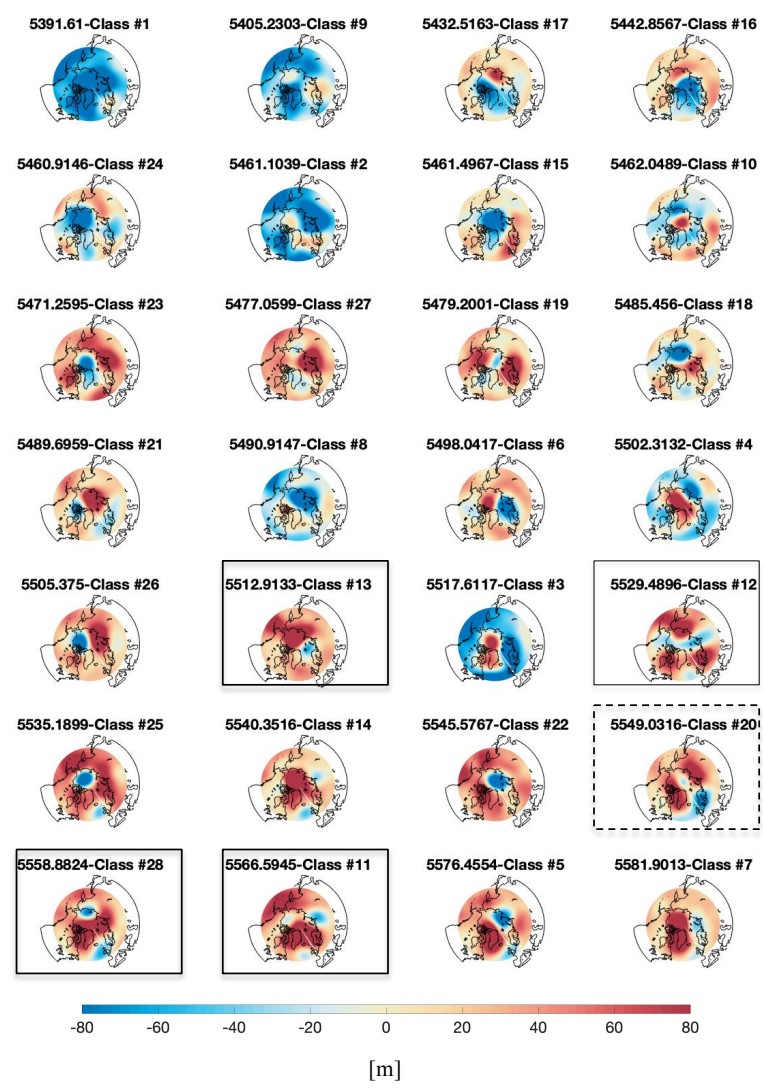

[m]

Figure 13 **Mean 5oo hPa GPH anomalies for the 28 classes identified by the SOM using the NCEP/NCAR reanalysis data for the period 1948 – 2019 ordered from the lowest to the highest mean GBI values computed using those days when the classes were occurring.**




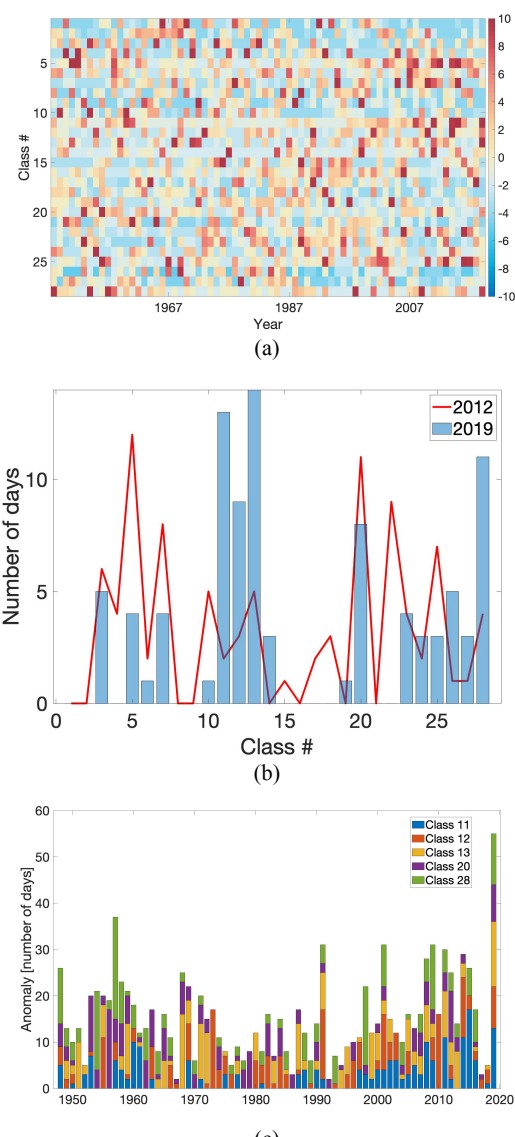

(a)

(b)


(c)

**Figure 14 a) Anomaly of the number of days (1981 – 2010 baseline) of the occurrence of each of the 28 classes identified through the SOM analysis for the years 1948 – 2019. B) Number of days when the 28 identified classes (x-axis) occur during the summers of 2012 (red line) and 2019 (blue bars). C) Anomaly of the cumulative number of**
**melting days for classes # 11,12,13,20 and 28 for the period 1948 – 2019.**