# Peer review of "Unprecedented atmospheric conditions (1948 – 2019) drive the 2019 exceptional melting season over the Greenland ice sheet"

_The Cryosphere, 2019_

## Short Comment (SC1) · 28 Nov 2019

Except figure 2b (blue line) shows the % of ice sheet surface was higher in 2012/2013, 2003/2004, and at the end of time series i.e 2019. Melting index was very low in 1980-1985, and during 2012-2017. It can be clearly shown the parameters (atmospheric/ice) those are exceptional in 2019 either in the form of time series graphs or maps. Otherwise, it's very good work. The manuscript is written well.

---

## Referee Comment (RC1) · Anonymous Referee #1 · 15 Jan 2020

This paper focuses on the atmospheric conditions in summer 2019 over Greenland that led to record or close-to-record values of SMB, runoff and snowfall. The topic is of great contemporary interest as these extreme melt events greatly impact the mass balance of Greenland and thus sea level rise. The authors are the first to present this kind of data for the year 2019.

Overall, I highly appreciate the rapid investigation of this recent event. The paper is well written but some small parts in the result section need clarification (see comments). The authors show very well the exceptional character of the 2019 melt season for

different properties (runoff, SMB, melt extent, etc).

However, as the authors write themselves, that anticyclonic conditions increase SMB loss is known so a more detailed comparison to the 2012 melt season and the melt seasons in general to post 1990 would improve this paper and add to our scientific understanding. Possibly there can be drawn some more conclusions.

Main comment:

For instance (see comments below): How does Figure 10 differ in the case for 2012, JJA or averaged post 1990, JJA. This anticyclone on the west of Greenland seems to be typically for the time post 1990, (Fig, 4 Noël et al., Sci. Adv. 2019) also prominent in the year 2012 (Tedesco, 2013). Noel et al. describe similar changes in cloud anomaly for the post 1990 as the authors did for the 2019 event. How does the 2019 event compare to that. Is it "just" more persistent or does it differ in position? How to the year 2012? The authors write that persistence was a major driver in 2019, so a comparison to the persistence of the other years would be beneficial (e.g. Plot of a timeline). Can we see a general trend in persistence there? How persistent was 2012?

Minor comments:

P3, L 92. What do you mean by that: The overall integrity of the long- term GBI time series is ensured by using homogeneity adjustments (Hanna et al. 2016). Please explain in the text.

P4, L l00. Since you refer to Fig2a, could you please either mention the day number in brackets for the dates in the text or change the axis label to actual dates.

P4, L100-102. Maybe make 2 sentences of this one? The explanation in the brackets is too lengthy and reduces the readability. And how do you come to the statements that the MI ranked 2nd . From Figure 2b red line it doesn't look like that. Or isn't it showing the MI? Please clarify.

P4, L 102-103. How do you see that? From Figure 2b I can only see the seasonal

value.

P4, L 108-110. I think the phrasing is a bit confusing and I needed to read the sentence several times. So are you saying that end of July 73% of the ice surface were melted and the following two days the remaining $\sim$ 23%? What do you mean by the same atmospheric conditions? Is the anticyclone moving or are the feedbacks increasing the melt area? Please clarify.

P4, L 111 -113. Please make a reference here.

P4. L113-114. You write the air masses in 2019 came from the east. For completeness, can you please add, where did the air masses in 2012 come from?

P4, L115-118. Interesting, but, -since it is in the result section- do you have the data or Figures showing that?

P6, L177- P7, L135. You say that persistence was a major driver in 2019.Was is more than in 2012? I like your definition of persistence showing a time line for 1948-2019 would be interesting. Can we see a trend?

Also how does Figure 10 differ in the case for 2012 or post 1990 JJA. This anticyclone on the west of Greenland seems to be typically for the time post 1990, (Fig, 4 Noël et al., Sci. Adv. 2019) also prominent in the year 2012 (Tedesco, 2013). It would be nice to see how the position of the anticyclone differs in respect to post 1990 or 2012.

Is it possible to check, whether the occurrence of any of the 28 classes correlated with the GBI anomaly or the SMB anomaly? Could you identify any significant trend?

Figures:

Fig 3a: Maybe put " $\leq$ 5" on the color bar, otherwise the reader might get the impression there are only 5 melting days.

Fig 14a: Could you please add a grid, otherwise the reader gets lost when searching for a specific class in a specific year.

Typos:

P3, L l76. "." after "("

P3, L l77. check "); ."

P6, L167. Something went wrong in the sentence order.

References:

Noël et al., Rapid ablation zone expansion amplifies north Greenland mass loss, Science Advances, 5, doi: 10.1126/sciadv.aaw0123

---

## Referee Comment (RC2) · Anonymous Referee #2 · 22 Jan 2020

The authors show that summer 2019 was an exceptional melt season for Greenland, with record or near-record values in runoff, snowfall, and SMB. They find that summer 2019 was characterized by persistent anticyclonic conditions and melting was enhanced by melt-albedo feedback and warm air advection. Comparing 2019 to the extreme melt season of 2012, they show that although the two years have similar values of runoff and SMB, these exceptional conditions were driven by different atmospheric circulation patterns.

This study provides valuable insights into this latest Greenland melt season within the

context of long-term trends. Overall, the analysis is thorough and well done, and the manuscript is well written.

— Main Comments —

In the SOM analysis, I'm concerned that the persistency of atmospheric patterns is defined somewhat arbitrarily and the exceptional persistency of 2019 may be overstated. In lines 246-249, 257-259, and Fig. 14c, the cumulative number of days for the top 5 most frequent classes of 2019 is compared to to the cumulative number of days of these same classes in other years and is found to be much higher. But this seems to be an inevitable result by construction, since the top 5 classes vary from year to year, so we're comparing 2019's top 5 classes to lesser ranked classes in other years. If these 5 classes were all adjacent on the SOM map, and represented some broader category of circulation pattern, then I could see how grouping them together is physically meaningful for comparison across years. But in this case, the grouping seems artificial and perhaps not a very robust approach to comparing across years.

For example, in Fig. 14c, the cumulative number of days for the 5 classes appears to be approximately 25 in 2012, much lower than the 55 days for these classes in 2019. However, we can see from Fig. 14b that 2012 had high persistency in atmospheric patterns as well, just with different classes than those in 2019. From Fig. 14b, I estimate the cumulative number of days for 2012's top 5 classes is approximately 47, which is pretty close in magnitude to the 55 days for 2019's top 5 classes. Thus, I would conclude that both 2012 and 2019 probably had high persistency, contrary to the analysis presented here.

If you were to repeat the analysis but instead compute the cumulative number of days for each year based on the top 5 classes in that specific year, then how do the results change? Is the persistency of atmospheric conditions in 2019 as exceptional as stated? It would be interesting to see if both 2012 and 2019 stand out as exceptional in this approach.

— Minor Comments —

L21-22 "the total number of days with the most frequent atmospheric pattern that characterized the summer of 2019 was 5 standard deviations above the 1981 – 2010 mean": This seems misleading, since the number of days referred to here is the cumulative total for the top 5 most frequent patterns of 2019, not the single most frequent pattern.

L102-103 "When looking at the different summer months separately, the MI values in 2019 ranked 5th in June, 7th in July and 9th in August (Fig. 2b)": Fig. 2b only shows summer averages, not monthly values, so perhaps the reference to it should be removed?

L109-110 "was also responsible for the cumulative 3-day melt event": Perhaps change "the cumulative..." to "a cumulative...", because this 3-day melt event hasn't been previously introduced. It would also be helpful to mention the specific dates of this event.

L112-118: Is this analysis of air mass trajectories all from the current study and details are not shown here? In particular regarding the 2012 summer, it's not clear if the discussion is summarizing earlier studies which should be cited here, or if it's referring to current analysis.

L167-168: The switch to positive albedo anomalies is confusing here, since the rest of the discussion centers around negative albedo anomalies. Also, looks like a typo here "In June, only 23%... was 23%."

L188-189: The text refers to "geopotential height anomalies", but the values listed (5567 m, etc.) seem to be actual geopotential heights, rather than anomalies.

L197 "high pressure system centered near Summit over the whole 2019 summer (Fig. 5a)": I think Fig. 10b should be referenced here, rather than Fig. 5a.

L198: I think Fig. 5e should be referenced here, rather than Fig. 5d.

L205-206: Should the area integrated anomalies be reported in units of W rather than W/m2?

L218 "We classify the daily 500 hPa GPH": Should specify that it's the GPH anomalies that are being classified.

L220-221: This description of the training phase with "existing SOM nodes" seems a bit off. The SOM nodes are defined in an iterative process during training – they don't exist prior to training.

L218-221: Were the input data fields weighted to account for grid cell area variation at high latitudes (for example, as in Mioduszewski et al. 2016)?

L229-230: Can you explain in more detail how the 4x7 SOM shape was selected? Were any sensitivity tests performed to determine the impact of SOM size / aspect ratio and analyze error metrics?

L263 "frequency and occurrence of the atmosphere": What does this mean?

— Figures —

Figure 1b: The colour scheme is inconsistent with Fig. 1a, Fig. 3a, and Fig. 5a, which all use red for more melting days and blue for fewer melting days. I recommend reversing the colour scheme in Fig. 1b to be consistent with the others.

Figure 2b: The caption describes the blue line as "Summer-averaged melt extent", but this is not a summer-averaged quantity, is it? It looks like the blue line shows, for each summer, the overall area subject to at least one day of melting.

Figure 5: Captions for subplots (b)-(e) are mixed up (i.e., (b) is snowfall anomaly but caption says 2m temperature, etc.). It would also be helpful to add a bit more horizontal space between subplots, so that there is some space between the colour bar labels and the y-axis of the right-adjacent subplot.

Figure 9: The annotation reads "Melt extent reaches ∼97%", whereas the main text

reports this value as 96%.

Figure 10a: In the caption "Anomaly of the JJA 2019 averaged geopotential height anomalies", extra "anomaly" should be removed.

Figure 11: Typo in the y-axis label: "yStandardized".

Figure 13: The subplots are very tiny. Can these be enlarged? Also, 3-4 decimal places in the average geopotential heights seems excessive - they could be rounded to 0 or 1 decimal place in these annotations.

Figure 14c: Caption and y-axis label describe this data as anomalies in the cumulative number of melting days, but the values shown aren't anomalies. Also, are they the cumulative number of melting days, or just cumulative number of days (melting or not)?

— Typographic Corrections —

Punctuation / spacing typos:

- L45 "i.e. ,Kohonen"

- L76 ".eg. Fettweis"

- L77 "2011); ."

- L90 "2015 ,2018;"

L115 "relative cold" ==> "relatively cold"

L259-260 "Despite similar in terms of runoff and SMB" ==> "Despite being similar. . ."

---

## Author Comment (AC3) · 13 Feb 2020

We thank the colleague for the comment
* * *

---

## Author Response (AR1)

qiangwu1970@gmail.com Received and published: 28 November 2019

Except figure 2b (blue line) shows the % of ice sheet surface was higher in 2012/2013, 2003/2004, and at the end
of time series i.e 2019. Melting index was very low in 1980- 1985, and during 2012-2017. It can be clearly shown the parameters (atmospheric/ice) those are exceptional in 2019 either in the form of time series graphs or maps. Other- wise, it's very good work. The manuscript is written well.

1

R: We thank the colleague for the comment

**Anonymous Referee #1**

Received and published: 15 January 2020

This paper focuses on the atmospheric conditions in summer 2019 over Greenland that led to record or close-torecord values of SMB, runoff and snowfall. The topic is of great contemporary interest as these extreme melt events greatly impact the mass balance of Greenland and thus sea level rise. The authors are the first to present this kind of data for the year 2019.

Overall, I highly appreciate the rapid investigation of this recent event. The paper is well written but some small parts in the result section need clarification (see comments). The authors show very well the exceptional

20 character of the 2019 melt season for different properties (runoff, SMB, melt extent, etc).

However, as the authors write themselves, that anticyclonic conditions increase SMB loss is known so a more detailed comparison to the 2012 melt season and the melt seasons in general to post 1990 would improve this paper and add to our scientific understanding. Possibly there can be drawn some more conclusions.

Main comment:

- 25 For instance (see comments below): How does Figure 10 differ in the case for 2012, JJA or averaged post 1990, JJA. This anticyclone on the west of Greenland seems to be typically for the time post 1990, (Fig, 4 Noël et al., Sci. Adv. 2019) also prominent in the year 2012 (Tedesco, 2013). Noel et al. describe similar changes in cloud anomaly for the post 1990 as the authors did for the 2019 event. How does the 2019 event compare to that. Is it "just" more persistent or does it differ in position? How to the year 2012? The authors write that persistence was
- 30 a major driver in 2019, so a comparison to the persistence of the other years would be beneficial (e.g. Plot of a timeline). Can we see a general trend in persistence there? How persistent was 2012?

*R*: We have replied to this point below when addressing the comments. It is not clear, also, to us what the reviewer means with a "trend" of persistence. We apologize for this and would be more than glad to address this point once clarified.

35 Minor comments:

P3, L 92. What do you mean by that: The overall integrity of the long- term GBI time series is ensured by using homogeneity adjustments (Hanna et al. 2016). Please explain in the text

In Hanna et al. (2016), the reanalysis 20CRv2 is used in addition of the NCEP-NCARv1 reanalysis to provide an "homogenous" GBI time series over the whole 20th century by combining 20CRv2 to NCEP-NCARv1 with the help of some

40 adjustments. Here, we discuss only the NCEP-NCARv1 based GBI index from 1948 (which is the JJA mean Z500 over 60-80°N, 20-80°W) and therefore this sentence is out of context here and we deleted it in the revised version of our manuscript.

P4, L 100. Since you refer to Fig2a, could you please either mention the day number in brackets for the dates in the text or change the axis label to actual dates.

**45 R: We have added the corresponding day of the year**

P4, L100-102. Maybe make 2 sentences of this one? The explanation in the brackets is too lengthy and reduces the readability. And how do you come to the statements that the MI ranked 2nd . From Figure 2b red line it doesn't look like that. Or isn't it showing the MI? Please clarify.

*R*: We have split the sentence and have corrected the fact that indeed 2019 ranks  $3^{rd}$ . thanks for catching this up.

50 P4, L 102-103. How do you see that? From Figure 2b I can only see the seasonal value.

**R: The reference to Fig2b was wrong. Thanks for catching that up.**

P4, L 108-110. I think the phrasing is a bit confusing and I needed to read the sentence several times. So are you saying that end of July 73% of the ice surface were melted and the following two days the remaining ~ 23%? What do you mean by the same atmospheric conditions? Is the anticyclone moving or are the feedbacks increasing the melt area? Please clarify.

**R*: we rewrote the sentence as follows:**

Indeed, the persistency of the atmospheric conditions at the end of July that were responsible for promoting melting over 73 % of the ice sheet in a single day (July 31, 2019) was extending melting during the next few days over regions that were not originally involved in the melting on July 31 with cumulative melt extent for the 3-day period (July 31 – August 2) reaching up to  $\sim$  97% of the ice sheet surface.

P4, L 111 -113. Please make a reference here.

R: Done, thanks !

55

60

P4. L113-114. You write the air masses in 2019 came from the east. For completeness, can you please add, where did the air masses in 2012 come from?

**65 *R: We mention this in the preceding sentence.**

P4, L115-118. Interesting, but, -since it is in the result section- do you have the data or Figures showing that?

Bellow, we can see the absolute values of the temperature at 700hPa (T700), 500hPa (T500) and mean specific humidity over 700-500hPa from NCEP-NCARv1 reanalysis on the 12th of July 2012 and on the 31st of July 2019. While the temperature anomalies were higher on 31-Jul-2019 than on 12-Jul-2019 with respect to the climatology of Mid-July or of the

70 end of July, the absolute values were higher in 2012 than in 2019. In addition, we can see that the humidity content was also higher in 2012 than in 2019 over the ice sheet, showing the important role of liquid clouds in the 2012 extreme melt event

(Bennartz et al., 2013). On this figure, we can finally see that the hot air bubble was centred over the ice sheet in 2012 but was rather along the north-east coast in 2019. These differences in temperature and humidity pattern explain why the 2012 highest melt event was more extreme than the 2019 one.

75 Bennartz, R., Shupe, M., Turner, D. et al. July 2012 Greenland melt extent enhanced by low-level liquid clouds. Nature 496, 83–86 (2013). https://doi.org/10.1038/nature12002

P6, L177- P7, L135. You say that persistence was a major driver in 2019.Was is more than in 2012? I like your definition of persistence showing a time line for 1948-2019 would be interesting. Can we see a trend?

*R*: we apologize but this question is not clear to us. The persistency of the 5 top 2019 atmospheric patterns is shown in Fig. 14. We are not sure the trend of which variable the reviewer is asking. Apologies again but this is not clear to us.

- Also how does Figure 10 differ in the case for 2012 or post 1990 JJA. This anticyclone on the west of Greenland
- 85 seems to be typically for the time post 1990, (Fig, 4 Noël et al., Sci. Adv. 2019) also prominent in the year 2012 (Tedesco, 2013). It would be nice to see how the position of the anticyclone differs in respect to post 1990 or 2012.

*R*: We added a new figure (Fig. 10, shown above) to highlight the differences between the 2012 and 2019 years and added the following text:

- As a reference, Fig. 11 shows the absolute values of the temperature at 700hPa (T700), 500hPa (T500) and mean specific humidity over 700-500hPa from NCEP-NCARv1 reanalysis on the July 12th 2012 and on the July 31st 2019. While the temperature anomalies were higher in 2019 with respect to the climatology of Mid-July or of the end of July, the absolute values were higher in 2012 than in 2019. In addition, the humidity content was also higher in 2012 than in 2019 over the ice sheet, showing the important role of liquid clouds in the 2012 extreme melt event (Bennartz et al., 2013). These differences in temperature and humidity pattern explain why the 2012 highest melt event was more extreme than the 2019 one.
  - Is it possible to check, whether the occurrence of any of the 28 classes correlated with the GBI anomaly or the SMB anomaly? Could you identify any significant trend?

*R*: This is a very interesting point and we did, indeed, look at this aspect. We found a modest correlation between the GBI and the frequency of SOMs nodes characterized by strong anticyclonic conditions. This is expected in

- 100 view of the definition of GBI. The strength of the correlation can change year by year. We think that this (together with the "modest" correlation) is due to the fact hat the GBI is an integrated measure of GPH anomalies over a specific geographic region where the atmospheric patterns identified with the SOMs extend well beyond that region. When we looked at the correlation between GBI / SOM frequency and melting we found similar correlations (summer integrated values). We are not sure if the reviewer is asking for trends in the GBI or
- 105 other quantities.

Figures:

Fig 3a: Maybe put " $\leq$  5" on the color bar, otherwise the reader might get the impression there are only 5 melting days.

R: Done

110 Fig 14a: Could you please add a grid, otherwise the reader gets lost when searching for a specific class in a specific year.

*R*: we tried to add a grid but the figure became hard to read. As an alternative, we have increased the resolution of the x-axis to improve readability.

Typos:

115 P3, L l76. "." after "("

R: Done.

P3, L 177. check "); ."

R: Done

P6, L167. Something went wrong in the sentence order.

120 *R: thanks for that. We fixed the sentence,*

**References:**

130

Noël et al., Rapid ablation zone expansion amplifies north Greenland mass loss, Science Advances, 5, doi: 10.1126/sciadv.aaw0123

**Anonymous Referee #2**

125 Received and published: 22 January 2020

The authors show that summer 2019 was an exceptional melt season for Greenland, with record or near-record values in runoff, snowfall, and SMB. They find that summer 2019 was characterized by persistent anticyclonic conditions and melting was enhanced by melt-albedo feedback and warm air advection. Comparing 2019 to the extreme melt season of 2012, they show that although the two years have similar values of runoff and SMB, these exceptional conditions were driven by different atmospheric circulation patterns.

This study provides valuable insights into this latest Greenland melt season within the context of long-term trends. Overall, the analysis is thorough and well done, and the manuscript is well written.

- Main Comments -

In the SOM analysis, I'm concerned that the persistency of atmospheric patterns is defined somewhat arbitrarily and the exceptional persistency of 2019 may be overstated. In lines 246-249, 257-259, and Fig. 14c, the cumulative number of days for the top 5 most frequent classes of 2019 is compared to to the cumulative number of days of these same classes in other years and is found to be much higher. But this seems to be an inevitable result by construction, since the top 5 classes vary from year to year, so we're comparing 2019's top 5 classes to lesser ranked classes in other years. If these 5 classes were all adjacent on the SOM map, and represented some

140 broader category of circulation pattern, then I could see how grouping them together is phys- ically meaningful for comparison across years. But in this case, the grouping seems artificial and perhaps not a very robust approach to comparing across years.

For example, in Fig. 14c, the cumulative number of days for the 5 classes appears to be approximately 25 in 2012, much lower than the 55 days for these classes in 2019. However, we can see from Fig. 14b that 2012 had

145 high persistency in atmospheric pat- terns as well, just with different classes than those in 2019. From Fig. 14b, I estimate the cumulative number of days for 2012's top 5 classes is approximately 47, which is pretty close in magnitude to the 55 days for 2019's top 5 classes. Thus, I would conclude that both 2012 and 2019 probably had high persistency, contrary to the analysis presented here.

If you were to repeat the analysis but instead compute the cumulative number of days for each year based on the top 5 classes in that specific year, then how do the results change? Is the persistency of atmospheric conditions in 2019 as exceptional as stated? It would be interesting to see if both 2012 and 2019 stand out as exceptional in this approach.

*R*: thanks for the comment. We have added the following text at the end of the Results to highlight the similarities between the 2012 and 2019 summers:

Nevertheless, we observe from Fig. 14b that 2012 also had high persistency in atmospheric patterns though such patterns belong to different classes than those in 2019. For example, the cumulative number of days for 2012's top 5 classes is 47, being similar in magnitude to the 55 days for 2019's top 5 classes. Moreover, similarly to 2019, the top five classes in 2012 were all characterized by high GPH anomalies and strong anticyclonic conditions (though different in terms of spatial distribution of the GPH anomalies). In this regard both 2012 and 2019 can be assumed to be exceptional from an atmospheric point of view.

Concerning the other years: we performed the analysis suggested by the reviewer of looking at the top 5 (or even 4 or 6) patterns (in terms of frequency) and found that 2012 and 2019 weree the years with the highest occurrence of the top 5 patterns. Moreover, we looked at the top 5 nodes (e.g., patterns) for the remaining years

and used the mean GBI (as well as the cumulative sum) to quantify the anticyclonic conditions and found that,
also in this case, the top 5 patterns were not characterized by anticyclonic conditions as strong as the ones in
2012 and 2019. Said this, we do acknowledge that there is extra work that needs or can be carried out to identify
"extreme" atmospheric patterns or their persistency and we plan to carry out a deeper analysis in our future
work and are thankful to the reviewer for his/her suggestions.

- Minor Comments -

170 L21-22 "the total number of days with the most frequent atmospheric pattern that characterized the summer of 2019 was 5 standard deviations above the 1981 – 2010 mean": This seems misleading, since the number of days referred to here is the cumulative total for the top 5 most frequent patterns of 2019, not the single most frequent pattern.

R: We changed the sentence as follows: The analysis of the frequency of daily 500 hPa geopotential heights
 obtained from artificial neural networks shows that the total number of days with the five most frequent
 atmospheric patterns that characterized the summer of 2019 was 5 standard deviations above the 1981 – 2010
 mean, confirming the exceptional nature of the 2019 season over Greenland.

L102-103 "When looking at the different summer months separately, the MI values in 2019 ranked 5th in June, 7th in July and 9th in August (Fig. 2b)": Fig. 2b only shows summer averages, not monthly values, so perhaps the reference to it should be removed?

**R: this was removed. Thanks.**

L109-110 "was also responsible for the cumulative 3-day melt event": Perhaps change "the cumulative..." to "a cumulative...", because this 3-day melt event hasn't been previously introduced. It would also be helpful to mention the specific dates of this event.

185 R: Done, thanks.

L112-118: Is this analysis of air mass trajectories all from the current study and details are not shown here? In particular regarding the 2012 summer, it's not clear if the discussion is summarizing earlier studies which should be cited here, or if it's referring to current analysis.

*R: We have added a reference to the first sentence to clarify the fact that it referes to a published study. Thanks for the suggestion.*

L167-168: The switch to positive albedo anomalies is confusing here, since the rest of the discussion centers around negative albedo anomalies. Also, looks like a typo here "In June, only 23%... was 23%."

R: Corrected, thanks.

L188-189: The text refers to "geopotential height anomalies", but the values listed (5567 m, etc.) seem to be actual geopotential heights, rather than anomalies.

R: thanks for catching this up. We have corrected the error. It should be simply "geopotential heights".

L197 "high pressure system centered near Summit over the whole 2019 summer (Fig. 5a)": I think Fig. 10b should be referenced here, rather than Fig. 5a.

R: That is correct. We have added the reference to the right figure. Thanks.

200 L198: I think Fig. 5e should be referenced here, rather than Fig. 5d.

R: That is correct. Thanks.

L205-206: Should the area integrated anomalies be reported in units of W rather than W/m2?

*R:* We prefer to list our anomalies in W/m2 as using W only is very dependent of ice sheet mask/resolution used, which we want to avoid according to the recommendations of Fettweis et al. (2020):

Fettweis, X., Hofer, S., Krebs-Kanzow, U., Amory, C., Aoki, T., Berends, C. J., Born, A., Box, J. E., Delhasse, A., Fujita, K., Gierz, P., Goelzer, H., Hanna, E., Hashimoto, A., Huybrechts, P., Kapsch, M.-L., King, M. D., Kittel, C., Lang, C., Langen, P. L., Lenaerts, J. T. M., Liston, G. E., Lohmann, G., Mernild, S. H., Mikolajewicz, U., Modali, K., Mottram, R. H., Niwano, M., Noël, B., Ryan, J. C., Smith, A., Streffing, J., Tedesco, M., van de Berg, W. J., van den Broeke, M., van de Wal, R. S. W.,

210 van Kampenhout, L., Wilton, D., Wouters, B., Ziemen, F., and Zolles, T.: GrSMBMIP: Intercomparison of the modelled 1980–2012 surface mass balance over the Greenland Ice sheet, The Cryosphere Discuss., https://doi.org/10.5194/tc-2019-321, in review, 2020.

L218 "We classify the daily 500 hPa GPH": Should specify that it's the GPH anomalies that are being classified.

215 R: we added "anomalies" here and in the successive sentence.

L220-221: This description of the training phase with "existing SOM nodes" seems a bit off. The SOM nodes are defined in an iterative process during training – they don't exist prior to training.

*R*: we added more explanation on the fact that the nodes are initially random structure in which the data is allocated.

220 L218-221: Were the input data fields weighted to account for grid cell area variation at high latitudes (for example, as in Mioduszewski et al. 2016)?

R: Yes.

L229-230: Can you explain in more detail how the 4x7 SOM shape was selected? Were any sensitivity tests performed to determine the impact of SOM size / aspect ratio and analyze error metrics?

- 225 *R: As we mention in the text, the selection of the number of nodes and the architecture of the SOM does not have specific rules but mostly directions. A first aspect to be considered is the computational time, as the network will have to train a large number of points (e.g., daily data for the past 60 summers gridded over the Arctic). One aspect also to account for is the fact that the number of elements to be classified are roughly the same within each node. We based the selection of our architecture on these two considerations by testing several*
- 230 configurations and building on our experience published in Mioduszewski et al, 2016. Once the architecture was selected, we trained 50 different SOMs and compare their outputs to assess the effect of the random initialization on the 50 different runs. We found that the differences in classification for the multiple SOMs was less than 0.1 %. We also tested our configuration using ad-hoc prepared training datasets with know shapes and patterns and tested the selected configuration. Also in this case, the error in terms of classification was smaller than 0.1 %.
- 235 L263 "frequency and occurrence of the atmosphere": What does this mean?

R: Thanks for pointing that out. We rewrote the sentence as follows:

In the future, we plan to analyse how the frequency and occurrence of GPH anomalies has been changing at higher levels (e.g., 300 hPa, 100 hPa) to eventually quantify potential missing links between the stratosphere and the troposphere that might be responsible for the exceptional conditions.

240 — Figures —

Figure 1b: The colour scheme is inconsistent with Fig. 1a, Fig. 3a, and Fig. 5a, which all use red for more melting days and blue for fewer melting days. I recommend reversing the colour scheme in Fig. 1b to be consistent with the others.

**R: we have reversed the colorbar.**

245 Figure 2b: The caption describes the blue line as "Summer-averaged melt extent", but this is not a summeraveraged quantity, is it? It looks like the blue line shows, for each summer, the overall area subject to at least one day of melting.

R: that is correct. We have rewritten the caption.

Figure 5: Captions for subplots (b)-(e) are mixed up (i.e., (b) is snowfall anomaly but caption says 2m
temperature, etc.). It would also be helpful to add a bit more horizontal space between subplots, so that there is some space between the colour bar labels and the y-axis of the right-adjacent subplot.

*R*: we corrected the caption and added the space.

Figure 9: The annotation reads "Melt extent reaches ~97%", whereas the main text reports this value as 96%.

*R*: we corrected the main text for the right value of 97 %.

255 Figure 10a: In the caption "Anomaly of the JJA 2019 averaged geopotential height anomalies", extra "anomaly" should be removed.

R: Done, thanks

Figure 11: Typo in the y-axis label: "yStandardized".

R: Adjusted, thanks

260 Figure 13: The subplots are very tiny. Can these be enlarged? Also, 3-4 decimal places in the average geopotential heights seems excessive - they could be rounded to 0 or 1 decimal place in these annotations.

*R*: We have removed the decimal place and enlarged the maps.

Figure 14c: Caption and y-axis label describe this data as anomalies in the cumulative number of melting days, but the values shown aren't anomalies. Also, are they the cumulative number of melting days, or just cumulative number of days (melting or not)?

*R*: thanks for this. We changed the caption accordingly.

- Typographic Corrections Punctuation / spacing typos:
- L45 "i.e. ,Kohonen"
- L76 ".eg. Fettweis"
- 270 L77 "2011); ."`
  - L90 "2015 ,2018;" L115 "relative cold" ==> "relatively cold"

[revised manuscript text omitted]

Marco T 2/11/20 05:07 Deleted: was ~ 23 %.

Marco T 2/12/20 09:20

| Formatted: Normal, Indent: First line: 0" |
|-------------------------------------------|
| Marco T 2/12/20 09:11                     |
| Deleted: 0                                |
| Marco T 2/12/20 09:11                     |
| Deleted: 0                                |
| Marco T 2/12/20 09:18                     |
| Deleted:                                  |
| Marco T 2/12/20 09:11                     |
| Deleted: 0                                |
| Marco T 2/12/20 09:19                     |
| Formatted: Font:Bold                      |
| Marco T 2/12/20 09:19                     |
| Formatted: Font:Not Bold, English (US)    |
| Marco T 2/12/20 09:19                     |
| Formatted: Font:Not Bold, English (US)    |
| Marco T 2/12/20 09:19                     |
| Formatted: English (US), Superscript      |
| Marco T 2/12/20 09:19                     |
| Formatted: Font:Not Bold, English (US)    |
| Marco T 2/12/20 09:20                     |
| Formatted: English (US), Superscript      |
| Marco T 2/12/20 09:19                     |
| Formatted: Font:Not Bold, English (US)    |

- 490 temperature and humidity pattern explain why the 2012 highest melt event was more extreme than the 2019 one. Over the center of the ice sheet, surface temperature was close to the 1981 - 2010 average, suggesting a larger role of the radiative forcing than the thermal one. The mean summer sea level pressure (SLP) averaged over the 60-80°N, 20-80°W region (i.e., the same area used to compute GBI, Hanna et al., 2016), reached a breaking record value of 1016 hPa vs. a 1981 - 2010 summer average of 1010+/-2 hPa. Also the summer averaged 500 hPa geopotential heights, integrated over the same area,
- set a new record of 5567 m, against a 1981 2010 average of 5497 +/- 25m (Fig. 12a). We computed the persistency of 495 anticyclonic conditions, defined here as the number of days when the daily mean SLP averaged over the Greenland ice sheet exceeds 1013 hPa (the common value of the standard pressure), and found that during the summer of 2019 such conditions existed for 63 of the 92 summer days (68 % of the summer). In perspective, the average number of days with the same conditions during the period 1981 - 2010 was 28+/- 12 days.
- 500 The anticyclonic conditions that characterized the summer of 2019 promoted negative cloudiness anomalies over the southern portion of the ice sheet and positive ones over the northern region (Fig. 5d), pointing to the important role of clouds in enhancing melting in this area (i.e., Hofer et al. 2017). In the North, the exceptional persistence of a high pressure system centered near Summit over the whole 2019 summer (Fig. 11h) favored advection of warm and wet air along the west
- side of Greenland towards the North, promoting higher than average surface temperatures (Fig. 5) and positive anomalies 505 of long wave downwelling radiation (Fig. 5f). In the southwest, dry and sunny conditions dominated. This promoted positive anomalies of the incoming shortwave radiation (Fig. 5g) which, in turn, when combined with the relatively low albedo (due to reduced summer snowfall) promoted positive anomalies of the absorbed shortwave radiation (Fig. 5h) higher than 30 W m2. Such drier conditions also allowed temperatures to reduce during nighttime, explaining why the temperature anomaly was not playing a larger role over these regions. Integrated over the whole ice sheet, the anomalies of shortwave and long
- wave downwelling radiation were not significant but, as a result of a quasi permanence of exposure of low albedo zones, the 510 anomaly of absorbed shortwave was the highest since 1948, with an anomaly integrated over the whole ice sheet of 7.9 W m 2, being four times the 1981-2010 standard deviation (inter-annual variability) of 1.9 W m2. The strong relationship between runoff and atmospheric conditions is also apparent in Fig. 13, where scatter plots of runoff with 500 hPa GPH summer mean anomalies (Fig. 13a) and with 700 hPa temperature (Fig. 13b) are shown, together with the coefficients of the linear
- 515 regression between runoff and the two atmospheric quantities. Reinforcing the idea that radiative forcing played a large role with respect to thermal forcing, the summer of 2019 (marked in the two panels with a large, orange circle) is beyond two standard deviations from the mean in the case of the 700 hPa temperature where it falls closely to the regression line in the case of the 500 hPa GPH.

To further understand the role of the atmosphere on the 2019 SMB record and the linkages between atmospheric 520 circulation and SMB, we classified summer (JJA) daily 500 hPa GPH anomalies between 1948 and 2019 into a set number of classes to study how the frequency of such classes has changed over the past decades and how the 2019 summer positioned itself within the 1948 - 2019 record. We focus on the 500 hPa GPH because of its strong correlation with the surface melt (Fettweis et al., 2011b) and because it is a standard height for gauging the effects of jet stream blocking on

rco T 2/11/20 05: Deleted: anomalies Marco T 2/12/20 09:11 Deleted: 1

| - | Marco T 2/11/20 05:18 |
|---|-----------------------|
|   | Deleted: 5            |
|   | Marco T 2/11/20 05:19 |
|   | Deleted: a            |
|   | Marco T 2/11/20 05:21 |
|   | Deleted: d            |

Marco T 2/12/20 09:1 Deleted: 2 Marco T 2/12/20 09:11 Deleted: 2 Marco T 2/12/20 09:11 Deleted: 2

[revised manuscript text omitted]